# Basic Pharmacological Characterization of EV-34, a New H_2_S-Releasing Ibuprofen Derivative

**DOI:** 10.3390/molecules26030599

**Published:** 2021-01-24

**Authors:** Alexandra Gyöngyösi, Vivien Verner, Ilona Bereczki, Attila Kiss-Szikszai, Rita Zilinyi, Árpád Tósaki, István Bak, Anikó Borbás, Pál Herczegh, István Lekli

**Affiliations:** 1Department of Pharmacology, Faculty of Pharmacy, University of Debrecen, 4032 Debrecen, Hungary; gyongyosi.alexandra@pharm.unideb.hu (A.G.); vivienverner@gmail.com (V.V.); zilinyi.rita@pharm.unideb.hu (R.Z.); tosaki.arpad@pharm.unideb.hu (Á.T.); 2Department of Pharmaceutical Chemistry, Faculty of Pharmacy, University of Debrecen, 4032 Debrecen, Hungary; bereczki.ilona@pharm.unideb.hu (I.B.); borbas.aniko@science.unideb.hu (A.B.); herczeghp@gmail.com (P.H.); 3Department of Organic Chemistry, Faculty of Science and Technology, University of Debrecen, 4032 Debrecen, Hungary; kiss.attila@science.unideb.hu; 4Department of Bioanalytical Chemistry, Faculty of Pharmacy, University of Debrecen, 4032 Debrecen, Hungary; bak.istvan@pharm.unideb.hu

**Keywords:** gasotransmitter, hydrogen-sulfide, H_2_S-releasing-ibuprofen, thiolacetic acid, anti-inflammatory property

## Abstract

Background: Cardioprotective effects of H_2_S are being suggested by numerous studies. Furthermore, H_2_S plays a role in relaxation of vascular smooth muscle, protects against oxidative stress, and modulates inflammation. Long-term high-dose use of NSAIDs, such as ibuprofen, have been associated with enhanced cardiovascular risk. The goal of the present work is the synthesis and basic pharmacological characterization of a newly designed H_2_S-releasing ibuprofen derivative. Methods: Following the synthesis of EV-34, a new H_2_S-releasing derivative of ibuprofen, oxidative stability assays were performed (Fenton and porphyrin assays). Furthermore, stability of the molecule was studied in rat serum and liver lysates. H_2_S-releasing ability of the EC-34 was studied with a hydrogen sulfide sensor. MTT (3-(4,5-dimethylthiazol 2-yl)-2,5-(diphenyltetrazolium bromide)) assay was carried out to monitor the possible cytotoxic effect of the compound. Cyclooxygenase (COX) inhibitory property of EV-34 was also evaluated. Carrageenan assay was carried out to compare the anti-inflammatory effect of EV-34 to ibuprofen in rat paws. Results: The results revealed that the molecule is stable under oxidative condition of Fenton reaction. However, EV-34 undergoes biodegradation in rat serum and liver lysates. In cell culture medium H_2_S is being released from EV-34. No cytotoxic effect was observed at concentrations of 10, 100, 500 µM. The COX-1 and COX-2 inhibitory effects of the molecule are comparable to those of ibuprofen. Furthermore, based on the carrageenan assay, EV-34 exhibits the same anti-inflammatory effect to that of equimolar amount of ibuprofen (100 mg/bwkg). Conclusion: The results indicate that EV-34 is a safe H_2_S releasing ibuprofen derivative bearing anti-inflammatory properties.

## 1. Introduction

Nonsteroidal anti-inflammatory drugs (NSAIDs) are the most commonly prescribed medications worldwide. However, they have several serious, potentially life-threatening adverse drug reactions [1]. During therapy with NSAIDs, patients are at risk of gastrointestinal and renal toxicity [2], which have long been known. Furthermore, an increase in arterial blood pressure upon administration of NSAIDs and the risk of exacerbation of heart failure have also been reported [1,3]. Cyclooxygenases catalyze the first step in the synthesis of prostanoids such as prostaglandin (PG) E2, PGD2, PGF2α, PGI2, and thromboxane (TxA2)—from arachidonic acid. Prostaglandins have an important role in the production of pain, inflammation, and fever [4]. Ibuprofen is one of the most commonly used NSAIDs, and it is an essentially non-selective inhibitor of cyclooxygenase-1 (COX-1) and cyclooxygenase-2 (COX-2). Ibuprofen at lower doses (below 1200 mg/day) possesses analgesic and antipyretic effect and at higher doses (above 2400 mg/day) has anti-inflammatory activities. Ibuprofen is strongly bound to plasma proteins and extensively metabolized in the liver by CYP enzymes. The major metabolites are hydroxylated and carboxylated compounds, excretion occurs via kidneys [5]. Ibuprofen possesses a relatively acceptable side-effect profile. However, long-term administration of higher dose (600–800 mg three times a day) of ibuprofen has enhanced the risk of adverse cardiovascular events and stroke [3,6].

In the past decades, hydrogen sulfide (H_2_S) has gained acceptance as the third gaseous mediator in mammals alongside nitric oxide (NO) and carbon monoxide (CO) [7]. One of the first described physiological effects of hydrogen sulfide was its ability to relax vascular smooth muscles, leading to vasodilation [8,9]. There is an increasing number of evidence supporting that H_2_S possesses activity on the circulatory system causing dilation of blood vessels, because H_2_S functions as an endothelial-derived hyperpolarizing factor (EDHF) and its vasorelaxant activity is associated with K_ATP_-channels [10]. However, an accumulating number of evidence suggests that the biological effect of H_2_S is not limited to a single pathway and a tissue type. It modulates different cellular targets in a cell and tissue dependent manner [11]. H_2_S can interfere with central metals, it can act as antioxidant coping with ROS/RNS and via S-persulfidation, it influences the function of different proteins [12]. 

As an example, H_2_S possess neuroprotective and cardioprotective effects [13,14,15]. Furthermore, it has been suggested that downregulated endogen H_2_S level plays a role in atherosclerosis; and exogenous H_2_S has therapeutic value against atherosclerosis via different mechanisms such as suppressing oxidative stress and inflammation, reducing endothelial dysfunction, platelet aggregation, and regulating lipid metabolism [16]. Furthermore, reduced serum H_2_S levels have been observed in patients suffering from heart failure and coronary heart disease [17,18]. Moreover, several studies have subsequently highlighted the importance of H_2_S in inflammatory processes. The ability of H_2_S to reduce inflammation has been demonstrated in a variety of animal models, including kaolin/carrageenan-induced monoarthritis in rats [19] or ischemia–reperfusion injury in mice [20]. Recently, H_2_S-donors have been suggested to have potential clinical value in COVID-19 therapy [21]. 

In mammalian cells, biosynthetic and degradative pathways involved in H_2_S production and degradation are largely mediated by cystathionine β synthase (CBS), cystathionine-γ-lyase (CSE), 3-mercaptopyruvate sulfurtransferase (3-MST), ethylmalonic encephalopathy protein 1 (ETHE1), mitochondrial sulfide–quinone oxidoreductase (SQR), and cysteine dioxygenase (CDO) [22]. Early results have shown that endogenous H_2_S concentrations are in micromolar range. However, later, it has been shown that H_2_S concentration in mouse brain and liver is around 20 nM [23,24]. Thus, accumulated evidence suggests a role for H_2_S in physiological and pathophysiological states [14]. Earlier, H_2_S-releasing aspirin, naproxen, and diclofenac derivatives had been synthetized as H_2_S donors and found to possess better GI tolerability, moreover for naproxen a better CV profile [25,26]. 

Thus, it is quite rational to hypothesize that a H_2_S release may contribute to anti-inflammatory property of ibuprofen and decrease cardiovascular risk when long-term treatment of NSAID is required. Thus, the purpose of the current project was the synthesis and basic pharmacological characterization of a H_2_S-releasing ibuprofen derivative, called compound EV-34. We have designed a conceptually new H_2_S releasing derivative, a prodrug containing a relatively stable formaldehyde bis-acylal structural part, which supposedly can be hydrolyzed in physiological conditions by a specific esterase enzyme, releasing thiolacetic acid. This compound, according to Liu [27], can form hydrogen sulfide in a reaction cascade.

## 2. Results and Discussion

### 2.1. Design and Synthesis of EV-34

Recently, two publications have appeared reporting on H_2_S-releasing ibuprofen derivatives [28,29]. We have designed and synthesized a new type of H_2_S donor ibuprofen derivative EV-34 containing a labile formaldehyde *O*,*S*-acylal functionality (Figure 1). First, ibuprofen (**1**) was converted to acyl chloride derivative (**2**) [30] from which, upon an acylation reaction with hydroxymethyl thiolacetate (**3**) [31], the desired acetylthiomethyl ester EV-34 was obtained in an acceptable yield. 

### 2.2. Oxidative Stability Assays

After the synthesis, oxidative stability of EV-34 compound was assessed utilizing two biomimetic model systems such as Fenton and synthetic porphyrin reactions.

First, the oxidation of the EV-34 was performed by the classical Fenton reaction as a model of phase I biotransformation, since it is suitable for modelling the phase I metabolic processes. Based on the recorded chromatograms of the blank, the control and test reactions, it was assumed that EV-34 is stable under the applied experimental conditions. Retention time for EV-34 is at 8.3 min evidenced by the chromatogram of pure EV-34, as it can be seen on Figure 2 panel A. No change was observed after the reaction of oxidation compared to the control chromatogram.

In the second set of experiments, a synthetic porphyrin, Fe(III) meso-tetra(4-sulfonatophenyl)porphine chloride was used. This method is suitable to mimic CYP450 oxidation. The metabolites were analyzed by GC-MS spectra (Figure 2B) at its base peak at *m*/*z* 161.1. The results suggest that EV-34 is metabolized under this condition, since the peak at 8.3 min in the control sample (oxidation reaction was not initiated) is under the limit of detection in the sample. Since ibuprofen is being metabolized via CYP enzymes, our result may indicate that CYP enzymes could also play a role in EV-34 metabolism.

Finally, stability of EV-34 was investigated in rat serum and liver lysates. The contribution of rat blood and liver lysates to the EV-34 oxidative status was evaluated by GC-MS. As it is depicted in Figure 2 panel C, a significantly lower AUC of peak at 8.3 min was observed in the presence of serum or liver lysates. These results are quite expected since the structure of the molecule allow ester hydrolysis. Furthermore, the ibuprofen moiety may serve as a substrate for CYP enzymes. However, further specific studies are required to analyze specific enzymes playing a role in EV-34 metabolism. 

### 2.3. H_2_S Releasing

H_2_S release of EV-34 molecule was analyzed with a direct hydrogen sulfide sensor. As it is depicted in Figure 3, H_2_S concentration started to increase after initiating the experiments and reached a maximal value after 16 min. Since the molecule contains single Sulphur atom, the theoretically maximal amount of H_2_S is therefore equivalent with the initial amount of EV-34. Thus, our results indicate that esterase and other enzymes located in the medium are essential for degradation of EV-34. Thus, these results with rat serum and liver lysates indicate that EV-34 is being cleaved in the biological system and H_2_S is released. Possible mechanisms of H_2_S release are shown in Figure 4. In the compound EV-34, the active part is a formaldehyde *O*,*S*-acylal (highlighted in yellow), which, following hydrolysis under mild physiological conditions, can result in hemiacylal (**4**) and thiolacetic acid, and the latter compound, according to Liu and Orgel [27], can release H_2_S (Figure 4). 

### 2.4. Safety Evaluation of EV-34

To assess the direct cytotoxic effects of EV-34, MTT assays were performed. Cells were incubated with EV-34 or ibuprofen at different concentrations (10, 100, and 500 µM). Cell viability of the untreated group was 100% and 92.45 ± 3.15% for DMSO treated control. Cell viability of EV-34 and ibuprofen treated cells were comparable with the untreated and vehicle treated control value (Figure 5A). In EV-34 treated cells, the viability was 85.16 ± 2.59%; 83.69 ± 3.67%; 77.7 ± 4.56%, and 100 ± 1.91%; 95.6 ± 1.04%; 87 ± 6.15% in ibuprofen treated cells, respectively. The slight decrement in viability of treated (ibuprofen, EV-34) groups in comparison with DMSO treated cells were not significant. Data are expressed as the mean ± SEM via 6 individual experiments. * *p* < 0.05, IBU 10 µM vs. EV-34 10 µM. 

To evaluate the safety profile of the EV-34 hemolytic activity, further studies were carried out. In the positive control group, rat erythrocytes were treated with ultrapure water, which induced 100% hemolysis. The hemolytic activity of different amount of EV-34 or ibuprofen was significantly lower compared to the positive control group. It has to be noted that in all treated groups, the hemolytic activity remained under 7%, which is similar to vehicle treated control group. Only significant differences can be observed among higher EV-34 treated groups (EV-34 800 and EV-34 1000) and the group treated by ibuprofen (IBU 800), 4.76 ± 0.11% and 4.88 ± 0.03% compared to 6.94 ± 0.1%, respectively, in hemolytic activities (Figure 5B). Therefore, the conclusion that EV-34 compound is an equally safe derivate as ibuprofen, can be drawn. Data are expressed as the mean ± SEM via 6 individual experiments. *** *p* < 0.001, IBU 800 vs. EV-34 800 and IBU 800 vs. EV-34 1000.

### 2.5. Anti-Inflammatory Effects of EV-34

To investigate the anti-inflammatory effects of the EV-34, an in vitro cyclooxygenase inhibition assay and an in vivo carrageenan-induced inflammation tests were performed, as it is depicted in Figure 6. A. EV-34 has the same effect on COX-1 and COX-2 inhibition as ibuprofen. These results indicate that EV-34 possesses COX inhibitory properties. 

In vivo carrageenan-induced inflammation tests were performed to compare the anti-inflammatory activity of EV-34 and ibuprofen. Carrageenan assay is used as an acute model of inflammation to evaluate anti-inflammatory activity of active substances [32,33]. The paw volume was measured before carrageenan injection and 2 h and 3 h after the injection of carrageenan by using electronic digital calipers. In order to rule out the role of mechanical injury, the same experiment was performed with saline. In saline control, after 2 h of injection, approximately 10% and after 3 h around 2% increment in paw diameter were detected. As shown in Figure 5B, edema appeared in the paw 2 h after carrageenan injection, and the tissue diameter increased in a time-dependent manner (after 2 h: 76.18 ± 6.21% and after 3 h: 84.8 ± 6.8%). Paw edema volumes followed the same trend after treatment. However, with the EV-34 or ibuprofen injection, the paw tissue edema was significantly reduced compared to the vehicle treated control. EV-34 and ibuprofen inhibited the development of edema as paw enlargement were 47.4 ± 4.33 and 55.8 ± 7.32% for EV-34 and 44.1 ± 5.64 and 46.4 ± 5.03% for ibuprofen, respectively after 2 and 3 h. Data are expressed as the mean ± SEM via data of 9–12 animals/group; * *p* < 0.05, ** *p* < 0.01, *** *p* < 0.001 versus the vehicle treated carrageenan injected group. Thus, we can conclude that EV-34 possesses similar anti-inflammatory effect to ibuprofen under our experimental condition. However, further comparison between EV-34 and ibuprofen treatment needs to be performed to evaluate the effect of long-term treatment with EV-34 on chronic inflammation.

## 3. Conclusions

Taken together, results suggest that similarly to earlier studies in which other H_2_S-releasing NSAIDs were studied, the newly synthetized EV-34 is equally safe as ibuprofen. Our postulation was right: our new ibuprofen prodrug was stable in aqueous solution, but released hydrogen sulfide in the presence of lysate. Under physiological circumstances, it releases H_2_S. In addition, EV-34 has COX inhibitory and anti-inflammatory properties same to ibuprofen. According to our knowledge, EV-34 is the first H_2_S-releasing ibuprofen derivative containing formaldehyde *bis*-acylal as active part. Earlier, a recent clinical trial called PRECISION, has found no significant differences between two non-selective COX inhibitor, namely ibuprofen, naproxen, and COX-2 selective rofecoxib with regard to CV risk [34,35]. Evidence suggests a role for endogenous H_2_S in maintaining physiological functions of the heart [14]. Furthermore, exogenous H_2_S has been shown to possess cardioprotective effect in a model of I/R and heart failure. H_2_S-releasing NSAIDs have been shown to exhibit a better GI side effect profile. Furthermore, for example, H_2_S-releasing naproxen exhibit better anti-inflammatory property than naproxen [36]. Thus, EV-34 could be a useful tool in the treatment of patients with high cardiovascular risk and long-term ibuprofen treatment, since our newly synthetized molecule has NSAID properties and also H_2_S-releasing ability. However, further studies are required to investigate the long-term general and organ specific toxicity, and biological efficacy of EV-34. 

## 4. Materials and Methods 

### 4.1. Chemistry—General Information

TLC was performed on Kieselgel 60 F254 (Merck, Darmstadt, Germany) with detection either by immersing into ammonium molybdate-sulfuric acid solution followed by heating for detection. Flash column chromatography was performed using Silica gel 60 (Merck, Darmstadt, Germany, 0.040–0.063 mm). The ^1^H-NMR (400 MHz) and ^13^C-NMR (100 MHz) spectra were recorded with a Bruker DRX-400 spectrometer. Chemical shifts are referenced to Me_4_Si (0.00 ppm for ^1^H) and to the solvent residual signals. ESI-QTOF MS measurement was carried out on a maXis II UHR ESI-QTOF MS instrument (Bruker, Billerica, MA, USA), in positive ionization mode. Constant background correction was applied for the spectrum; the background was recorded before the sample by injecting the blank sample matrix (solvent). Na-formate calibrant was injected after the sample, which enabled internal calibration during data evaluation.

### 4.2. Synthesis of Ibuprofen Acetylthiomethyl Ester EV-34

Hydroxymethyl thiolacetate **3** (3.74 g, 35.25 mmol) was dissolved in 25 mL of abs. dichloromethane and triethylamine (7 mL, 74 mmol) was added. The reaction mixture was cooled to 0 °C and the solution of ibuprofen chloride **2** (5.27 g, 23.5 mmol) in 25 mL of abs. dichloromethane was added dropwise. The mixture was stirred for 24 h at room temperature, then washed with 2% citric acid solution and saturated NaHCO_3_ solution. After evaporation, the crude product was purified by flash column chromatography (hexane-ethylacetate 99:1) to yield EV-34 (2 g, 29%) as a yellowish syrup.

^1^H-NMR (400 MHz, CDCl_3_) δ 7.15 (m, 4H, aromatic), 5.38 (q, *J* = 10.9, 2H, CH_2_), 3.86 (q, *J* = 7.1 Hz, 1H, CH), 2.45 (t, *J* = 6.1 Hz, 2H, CH_2_), 2.03 (s, 3H, CH3), 1.96–1.73 (m, 1H, CH), 1.54 (d, *J* = 7.1 Hz, 3H, CH_3_), 0.90 (d, *J* = 6.6 Hz, 6H, CH_3_); ^13^C-NMR (100 MHz, CDCl_3_) δ 199.0 (1C, C=O), 170.5 (1C, C=O), 141.4, 136.2, 129.6, 127.9 (6C, aromatic), 61.2 (1C, CH_2_), 54.2 (1C, CH), 45.1 (1C, CH_2_), 30.2, 22.4, 20.8, 18.2 (5C, CH, CH_3_). ESI-TOF-MS: Calculated for C_16_H_22_O_3_S [M + Na]^+^: 317.1187, found: 317.1181.

### 4.3. Oxidation by the Chemical Fenton System and Synthetic Porphyrin 

Fenton reaction and synthetic metalloporphirin assay were carried to monitor oxidative stability. These two reactions were performed to test the stability of compound EV-34, based on the method reported previously by Csepanyi et al. [37]. Briefly, 1 mg of EV-34 dissolved in 1% DMSO was used for the Fenton reaction at 2.5 mM concentration and for synthetic porphyrin oxidation at 10 mM concentration. In case of Fenton reaction, each tube contained 20 mM FeCl_3_, 20 mM EDTA and 10 mM ascorbic acid. Reaction mixtures for blank contained 1% DMSO only without EV-34. The control tube contained all reagents and EV-34, except hydrogen peroxide, which is responsible for initiation of the reaction. Fenton reaction mixtures were stirred for 30 min at room temperature. Synthetic porphyrin reaction mixtures contained acetonitrile, 100 mM formic acid, 10 mM Fe(III)-meso-tetra(4-sulfonatophenyl) porphine, EV-34, and hydrogen peroxide. Then these mixtures were stirred for 30 min at 50 °C in Eppendorf ThermoMixer C (Eppendorf AG, Hamburg, Germany). The samples were analyzed by a GC-MS (Agilent 7890A GC coupled with an 5975C MSD, Agilent Technologies Ltd., Santa Clara, CA, USA).

### 4.4. Oxidative Stability in Rat Blood and Liver Lysates

To mimic biological environment, rat serum and liver lysates were used to study stability of EV-34. Rat blood samples were collected into CAT (Clot Activator Tube) Plus Blood Collection Tubes (BD Vacutainer, Plymouth, UK), and centrifuged. EV-34 in a concentration of 20 µM was added to the serum fraction. Liver tissue lysate was prepared by homogenization in RIPA buffer (50 mM Tris-HCl, pH 8.0, with 150 mM sodium chloride, 1.0% Igepal CA-630 (NP-40), 0.5% sodium deoxycholate, and 0.1% sodium dodecyl sulfate). Tissue and cell debris were removed by centrifugation. Liver lysates were treated with 20 µM of EV-34. Each sample with or without EV-34 was incubated for 20 min at 37 °C. After it, acetonitrile (1:2 ratio) was added and samples were centrifuged for 5 min, at 10,000× *g*. The resulted supernatants were analyzed immediately by a GC-MS/MS as described in the previous section.

### 4.5. H_2_S Releasing

Direct measurement of H_2_S was carried out with an amperometric H_2_S sensor (ISO-H2S-100 (World Precision Instruments, Sarasota, FL, USA) connected to a WPI TBR 1025 One-Channel Free Radical Analyzer. Briefly, H9c2 cells were cultured in Dulbecco’s modified Eagle’s medium (DMEM). After 3 days, the medium was collected and EV-34 dissolved in 1% DMSO was added at a final concentration of 50 µM. H_2_S liberation was detected continuously during 6 h at room temperature. Control experiments were carried out without EV-34.

### 4.6. Cell Culture and Treatment for Determination of Cytotoxicity by MTT Assay

H9c2 cells were cultured in DMEM supplemented with 10% fetal bovine serum and 1% streptomycin-penicillin at 37 °C in a humidified incubator consisting of 5% CO_2_ and 95% air. Cells were cultured for one day to establish adhesion of the wells. Confluent cells (60–70% confluence) were employed to the experiments. EV-34 was dissolved in 1% DMSO. All reagents were purchased from Sigma (St. Louis, MO, USA).

Cell viability was measured by MTT (3-(4,5-dimethylthiazol 2-yl)-2,5-(diphenyltetrazolium bromide)) experiments on 96-well plates. Cells were exposed to ibuprofen and EV-34 at the following concentrations: 10 μM, 100 μM, 500 μM for 24 h. The untreated control group was incubated only with medium, and the vehicle treated control group with medium containing 1% DMSO. After treatment, MTT solution (final concentration of 0.5 mg/mL) was added to each well and incubated for 3.5 h at 37 °C. Then, medium was replaced by isopropyl alcohol to dissolve formazan product and incubated for 30 min at 37 °C. Absorbance was measured with Multiskan GO Microplate Spectrophotometer (Thermo Fisher Scientific Oy, Ratastie, Finland) at 570 and 690 nm. The values were expressed relative to the untreated control, which was represented as 100% of viability. One percent H_2_O_2_ was used as positive control. Absorbance values were averaged across 5 replicate wells, and repeated 6 times.

### 4.7. Animals

Male Sprague Dawley (SD) rats (Charles River Laboratories International, Inc. Sulzfeld, Germany) with an average weight of 510 ± 12 g were used in the present study. All animals were housed and treated according to the “Principles of Laboratory Animal Care” formulated by the National Society for Medical Research and the “Guide for the Care and Use of Laboratory Animals” prepared by the National Academy of Sciences and published by the National Institutes of Health (NIH Publication no. 86–23, revised in 1996). Maintenance and treatment of animals taken part in this study were additionally approved by the Institutional Animal Care and Use Committee of the University of Debrecen, Debrecen, Hungary. Rats were housed in wire-bottomed cages maintained on 12:12-h light-dark cycle throughout the study and nutrified with standard rodent chew pellets ad libitum with free access to water. Approval number: 6/2019/DEMÁB.

### 4.8. Determination of Hemolytic Activity

A hemolysis test was carried out as described by Roka et al. with some minor modifications [38]. Rat blood samples were collected into 9NC 0.105M buffered sodium citrate vacuum tubes (BD Vacutainer, Plymouth, UK), centrifuged, and red blood cells were treated with 200 µM, 400 µM, 800 µM, and 1 mM of EV-34 or equimolar concentration of ibuprofen in phosphate buffered saline (PBS). Ultrapure water was used as positive control as it has been described earlier [39,40], while PBS served as the negative control. After mixing the samples gently, each solution was incubated at room temperature for 10 min, and then centrifuged at 5000× *g*. The absorbance of the hemoglobin released into the supernatant was measured at 540 nm with Multiskan GO Microplate Spectrophotometer (Thermo Fisher Scientific Oy, Ratastie, Finland). Hemolytic activity was expressed as a ratio of the treated group and positive control.

### 4.9. Carrageenan-Induced Inflammation Tests

The paw edema test was performed as described previously by Fehrenbacher et al. [32]. The left paw of each animal was injected with 100 μL of 1% carrageenan solution with the help of a sterile 1-mL syringe equipped with a 27-G,1/2-inch needle. Two minutes after, rats received a single intraperitoneal injection of ibuprofen-sodium (100 mg/kg) or EV-34 (similar to equimolar amount of ibuprofen), or saline solution in control animals. To rule out the effect of the volume load of the paw injection, three rats received saline solution to the left paw. Before carrageenan injection, baseline paw diameter was measured using electronic digital caliper. Following 2 and 3 h of carrageenan injection, paw diameter was re-measured and edema was calculated for each animal. The difference in the paw diameter before and after the injection was compared to the baseline diameter and used as the measurement of edema formation due to inflammation.

### 4.10. In Vitro Cyclooxygenase Inhibition Assay

The ability of EV-34 and ibuprofen to inhibit COX-1/COX-2 was determined using a colorimetric COX (human) inhibitor screening assay kit (Cayman Chemical, Ann Arbor, MI, USA, Catalogue No. 701230) as the protocol recommended by the supplier. 

### 4.11. Statistical Analyses

All data are presented as the average magnitudes of each outcome in a group ± standard error of the mean. Statistical analysis was performed using one-way analysis of variance (ANOVA), followed by Tukey’s multiple comparisons test with GraphPad Prism software for Windows (GraphPad Software Inc., LaJolla, CA, USA). Probability values (*p*) < 0.05 were considered statistically significant * *p* < 0.05, ** *p* < 0.01, *** *p* < 0.001.

## Figures and Tables

**Figure 1 molecules-26-00599-f001:**
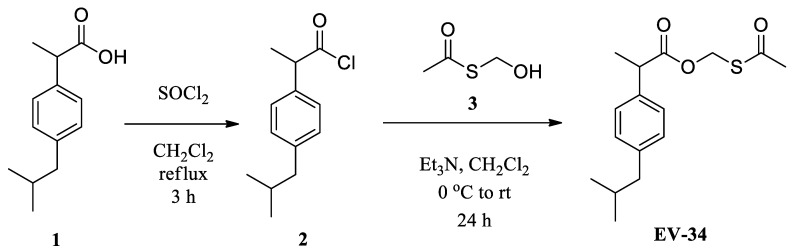
Synthesis of the new H_2_S-releasing derivative of ibuprofen.

**Figure 2 molecules-26-00599-f002:**
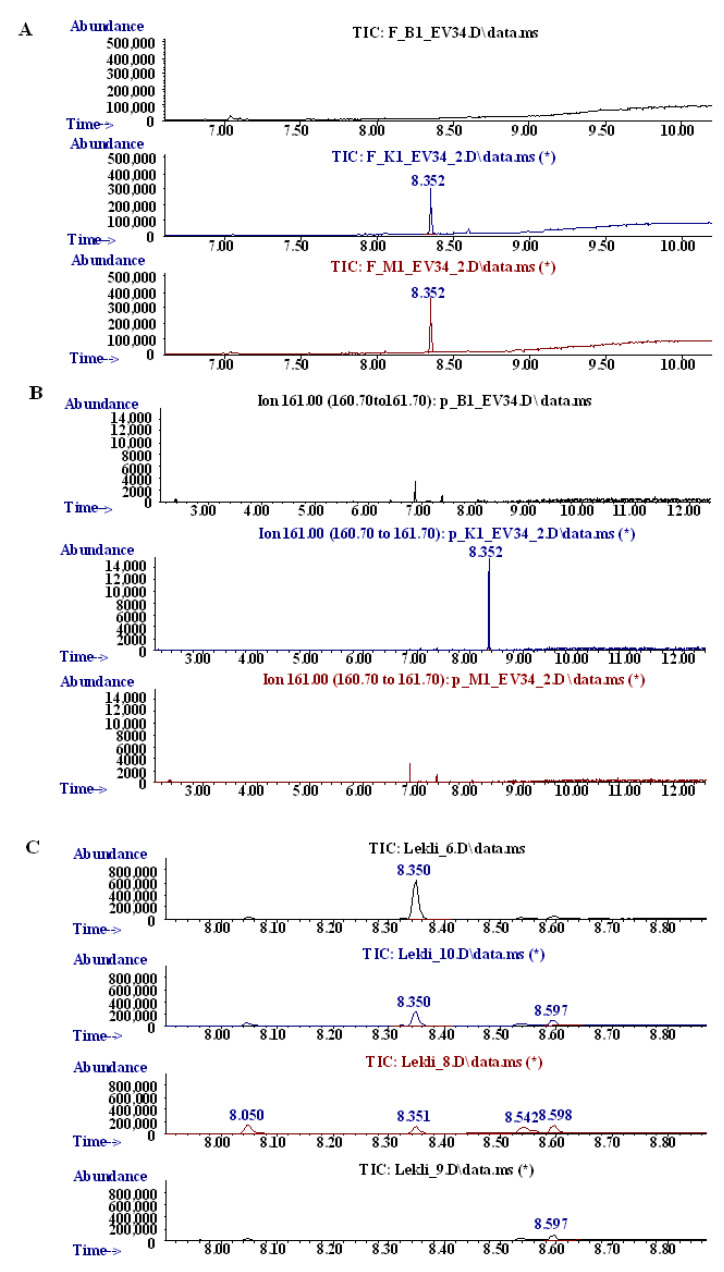
Stability of EV-34: (**A**) Oxidation by the chemical Fenton system. Reaction mixtures for blank contained 1% DMSO only without EV-34. The control tube contained all reagents and EV-34 except hydrogen peroxide, which is responsible for the initiation of the reaction. The sample tube contained all reagents; EV-34 concentration was 2.5 mM. (**B**) Mimic oxidation by synthetic porphyrin. The sample tube contained all reagents; EV-34 concentration was 10 mM. (**C**) To mimic biological environment stability of EV-34 was studied in rat blood and liver lysates. EV-34 concentration was 20 µM in the samples.

**Figure 3 molecules-26-00599-f003:**
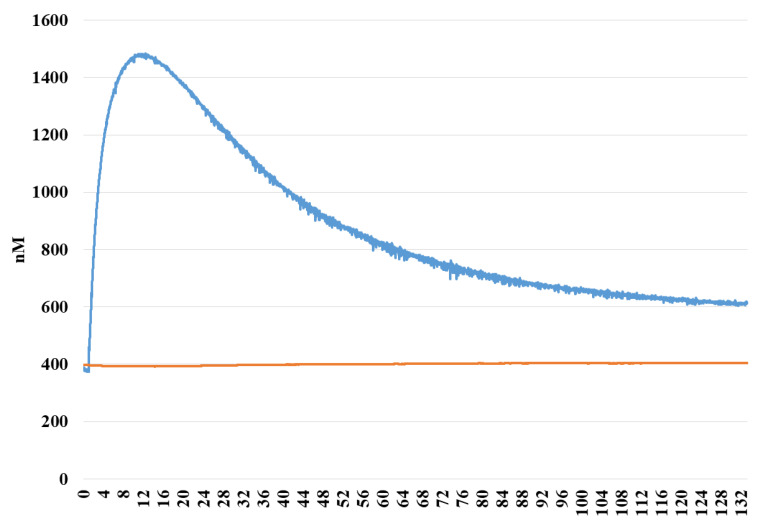
H_2_S-releasing of EV-34: Alteration in H_2_S concentration in media obtained from cells after 3 days of incubation. Blue: EV-34, orange: 1% DMSO.

**Figure 4 molecules-26-00599-f004:**
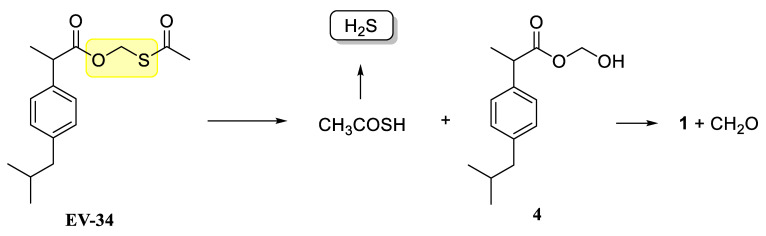
Possible reaction pathways for the release of H_2_S from EV-34.

**Figure 5 molecules-26-00599-f005:**
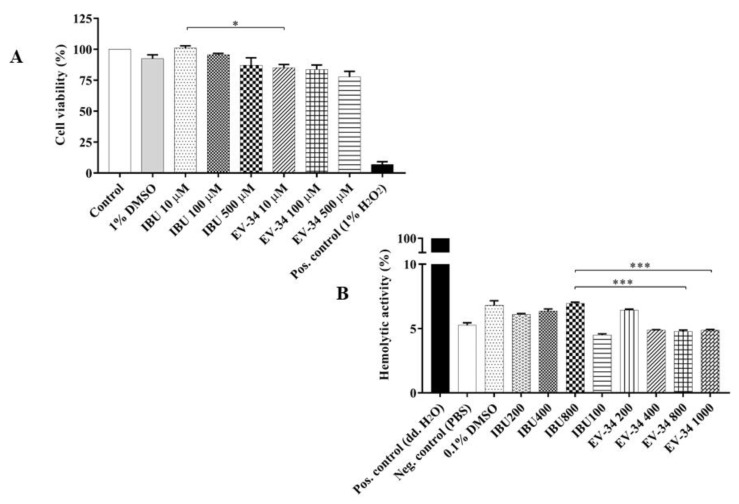
Safety evaluation of EV-34. (**A**) Cytotoxicity test. H9c2 cells were treated with various concentrations (10 μM, 100 μM, 500 μM for 24 h) of ibuprofen or EV-34. 1% DMSO was used as vehicle treated control. Cell viability was measured by MTT assay. Viability was reported as percentages of cell surviving ibuprofen or EV-34 exposure compared to the untreated group. The measurements were carried out in quintuplicate. Data are expressed as the mean ± SEM (*n* = 6); * *p* < 0.05. (**B**) Hemolytic activity of ibuprofen or EV-34 on rat red blood cells. Hemolysis was expressed as the percentage of untreated control, which contained the solvent (PBS) only. Positive control: purified water. Analyzed concentrations of ibuprofen or EV-34 were 200 µM, 400 µM, 800 µM, and 1 mM. Data are expressed as the mean ± SEM (*n* = 6); *** *p* < 0.001.

**Figure 6 molecules-26-00599-f006:**
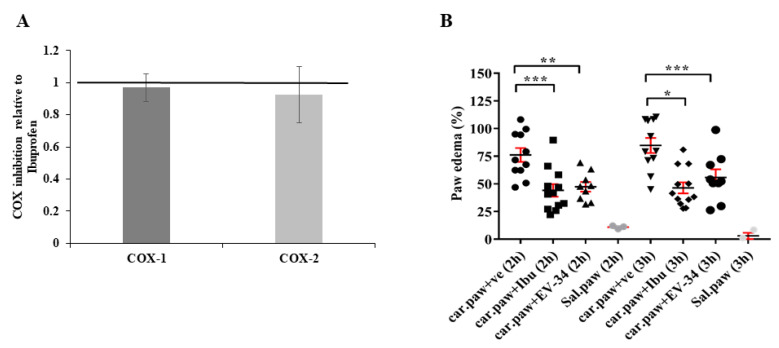
Anti-inflammatory effects of EV-34. (**A**) COX-1 and COX- inhibition relative to ibuprofen. (**B**) Effects of ibuprofen and EV-34 on carrageenan-induced edema in rat paws. The degree of edema in the rat paw tissue was calculated as the ratio of the change in paw volume between the basal volume (0 h) and at different time intervals of 2, 3 h after carrageenan injection. After intra-paw injection of carrageenan rats were treated intraperitoneally either with ibuprofen-sodium (100 mg/kg), or EV-34, or saline as vehicle control. Data are expressed as the mean ± SEM (*n* = 9–12); * *p* < 0.05, ** *p* < 0.01, *** *p* < 0.001 versus the vehicle treated carrageenan injected group. A total of 3 rats were given intra-paw injection of saline as volume load control.

## Data Availability

The data that support the findings of this study are available from the corresponding author, upon reasonable request.

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
