# Peer review of "Basic Pharmacological Characterization of EV-34, a New H2S-Releasing Ibuprofen Derivative"

_molecules, 2021, doi:10.3390/molecules26030599_

Round 1
Reviewer 1 Report
A simple but solid presentation of results supporting hypothesis.
In this manuscript, authors synthesized an ibuprofen derivative EV-34 designed to release H2S that would protect vascular smooth muscle from oxidative stress and inflammation. EV-34 was then tested for H2S release and anti-inflammatory activity. Synthesis and experimental design is straightforward and conclusion was adequately supported by experimental data. Therefore, the reviewer recommend acceptance of the manuscript for publication after minor revisions detailed below:
- Several overlaps seen in figure 2 should be fixed. For instance, TIC:F_B1_EV34 and data.ms are overlapped and not easy to read.
- Line 48: highly à tightly?
Minor correction in English desired in lines 48 and 57.
Author Response
Reviewer I.
First, we would like to thank for the suggestions and comments raised by this Reviewer, we tried to incorporate all of them in the revised version of our manuscript. We believe that the comments improve the quality of our manuscript. We responded point by point to all suggestions:
- Several overlaps seen in figure 2 should be fixed. For instance, TIC:F_B1_EV34 and data.ms are overlapped and not easy to read.
We have fixed the figure as requested by the reviewer.
- Line 48: highly à tightly?
We modified as „strongly”.
Minor correction in English desired in lines 48 and 57.
We asked an expert to check our manuscript.
Reviewer 2 Report
This is an interesting manuscript. I have following specific comments:
1) There are issues with visibility of elements in Figure 2. Please, amend this figure.
2) Define n in statistical analysis. When means are mentioned in the text, please add SEM plus n and p values.
Author Response
Reviewer 2
First, we would like to thank for the suggestions and comments raised by this Reviewer, we tried to incorporate all of them in the revised version of our manuscript. We believe that the comments improve the quality of our manuscript. We responded point by point to all suggestions:
- There are issues with visibility of elements in Figure 2. Please, amend this figure.
We have fixed the figure as requested by the reviewer.
- Define n in statistical analysis. When means are mentioned in the text, please add SEM plus n and p values.
We have included the requested values to the results.
Reviewer 3 Report
Abstract: What is the objective of synthesizing EV-34? Authors mention cardiovascular risk of ibuprofen, but their study does not include to evaluate whether EV-34 has no or less cardiovascular risk.
- 2, line 50: Wrong use of the word “favorable” because no side effect can be favorable.
- 2, line 60: Paragraph should be changed from the sentence starting with “Moreover”.
Line 83: Authors mention report of two ibuprofen derivatives in literature which can provide hydrogen sulfide and now they have synthesized a new one. Authors should have discussed the rationale for synthesizing EV-34. Was there any problem with the reported hydrogen sulfide releasing ibuprofen derivates?
Lines 143-149: All the % data should have standard deviations values too. The statement in lines 148-149 “The decrement …. significant” is not correct on prima facie as per Fig. 5A.
Lines 150-159: Selection of ultrapure water is not a good positive control because it is hypotonic, so obviously, it would cause hemolysis. Authors have concluded that EV 40 is non-toxic because it is causing less hemolysis than the positive control. This in turn would mean ultrapure water is toxic which is not scientifically valid use of word “toxic” or “non-toxic”.
Line 203: It is concluded that EV-34 is biocompatible, but authors have not conducted any experiment to evaluate biocompatibility of EV-34.
205-206: Authors claim that EV-34 is the first ibuprofen derivate releasing H2S but line 83 mentions two such derivatives reported in literature and they have cited 22 and 23. Both cannot be correct.
Author Response
Reviewer 3
First, we would like to thank for the suggestions and comments raised by this Reviewer, we tried to incorporate all of them in the revised version of our manuscript. We believe that the comments improve the quality of our manuscript. We responded point by point to all suggestions:
- 2, line 50: Wrong use of the word “favorable” because no side effect can be favorable.
We have changed the word „favorable” to „relatively acceptable”.
- 2, line 60: Paragraph should be changed from the sentence starting with “Moreover”.
We have rearranged the section as it was required and now it reads as follows:
„ However, accumulating number of evidence suggest that the biological effect of H2S is not limited to a single pathway and a tissue type. It modulates different cellular targets in a cell and tissue dependent manner [11]. H2S can interfere with central metals, it can act as antioxidant coping with ROS/RNS and via S-persulfidation, it influences the function of different proteins [12].
As an example, H2S possess neuroprotective and cardioprotective effects [13-15]. Furthermore, it has been suggested that downregulated endogen H2S level plays a role in atherosclerosis; and exogenous H2S has therapeutic value against atherosclerosis via different mechanisms such as suppressing oxidative stress and inflammation, reducing endothelial dysfunction, platelet aggregation, and regulating lipid metabolism [16]. Furthermore, reduced serum H2S levels have been observed in patients suffering from heart failure and coronary heart disease [17,18]. Moreover, several studies have subsequently highlighted the importance of H2S in inflammatory processes. The ability of H2S to reduce inflammation has been demonstrated in a variety of animal models, including kaolin/carrageenan-induced monoarthritis in rats [19] or ischemia–reperfusion injury in mice [20]. Recently, H2S-donors have been suggested to have potential clinical value in COVID-19 therapy [21].
In mammalian cells, biosynthetic and degradative pathways involved in H2S production and degradation are largely mediated by cystathionine β synthase (CBS), cystathionine-γ-lyase (CSE), 3-mercaptopyruvate sulfurtransferase (3-MST), ethylmalonic encephalopathy protein 1 (ETHE1), mitochondrial sulfide–quinone oxidoreductase (SQR), and cysteine dioxygenase (CDO) [22]. Early results have shown that endogenous H2S concentrations are in micromolar range. However, later it has been shown that H2S concentration in mouse brain and liver is around 20 nM [23,24]. Thus, accumulated evidence suggests a role for H2S in physiological and pathophysiological states [14].”
Line 83: Authors mention report of two ibuprofen derivatives in literature which can provide hydrogen sulfide and now they have synthesized a new one. Authors should have discussed the rationale for synthesizing EV-34. Was there any problem with the reported hydrogen sulfide releasing ibuprofen derivates?
The reviewer is right, there are few new H2S releasing ibuprofen derivatives. We added the following section to the introduction:
“We have designed a conceptually new H2S releasing derivative, a prodrug containing a relatively stable formaldehyde bis-acylal structural part, which supposedly can be hydrolyzed in physiological conditions by aspecific esterase enzymes, releasing thiolacetic acid. This compound, according to Liu [27] can form hydrogen sulfide in a reaction cascade.”
and the the following sentence to the conclusion:
“Our postulation was right: our new ibuprofen prodrug was stable in aqueous solution, but released hydrogen sulfide in the presence of lysate.”
Lines 143-149: All the % data should have standard deviations values too. The statement in lines 148-149 “The decrement …. significant” is not correct on prima facie as per Fig. 5A.
We added SEM values. We have modified the section as the follows:
“…was 100% and 92.45±3.15% for DMSO treated control. Cell viability of EV-34 and ibuprofen treated cells were comparable with the untreated and vehicle treated control value (Fig. 5.A). In EV-34 treated cells the viability was 85.16±2.59%; 83.69±3.67%; 77.7±4.56%, and 100±1.91%; 95.6±1.04%; 87±6.15% in ibuprofen treated cells, respectively. The slight decrement in viability of treated (ibuprofen, EV-34) groups in comparison with DMSO treated cells were not significant. Data are expressed as the mean ±SEM via 6 individual experiments. *p<0.05, IBU 10 µM vs, EV-34 10 µM.”
Lines 150-159: Selection of ultrapure water is not a good positive control because it is hypotonic, so obviously, it would cause hemolysis. Authors have concluded that EV 40 is non-toxic because it is causing less hemolysis than the positive control. This in turn would mean ultrapure water is toxic which is not scientifically valid use of word “toxic” or “non-toxic”.
It was not our aim to suggest that ultrapure water is toxic. Thus, we omitted the sentence from the section: „This result further confirms that our new derivative is a nontoxic compound under our experimental conditions.”
Furthermore, we modified the discussion as the follows. „the newly synthetized EV-34 is equally safe as ibuprofen.”
However, there are reports in which ultrapure water is being used as positive control in hemolysis assay. Thus, we added two references. ref.: 39,40
Line 203: It is concluded that EV-34 is biocompatible, but authors have not conducted any experiment to evaluate biocompatibility of EV-34.
We omitted the word „biocompatible”.
205-206: Authors claim that EV-34 is the first ibuprofen derivate releasing H2S but line 83 mentions two such derivatives reported in literature and they have cited 22 and 23. Both cannot be correct.
The reviewer is correct. Recently, few new H2S-releasing ibuprofen derivative were described. Thus. we modified the sentence as the follows:
„According to our knowledge EV-34 is the first H2S-releasing ibuprofen derivative containing formaldehyde bis-acylal as active part.”
Reviewer 4 Report
The authors report the activity of EC-34, which is a H2S releasing NSAID derivative. The authors report studies on H2S release, cytotoxicity, COX inhibition, and anti-inflammation models in rat paws. This is an interesting report, but a number of points need further clarification before publication.
a. The authors state that “Results of Olson [18] showed that endogenous H2S concentrations are between 30 and 300 μM in the plasma and blood” and reference a paper over a decade old. These high levels of H2S in plasma and blood are incorrect, and the contemporary literature agrees that free H2S levels are below 1 micromolar. The authors need to update this introductory material. (As a quick check- the human nose can smell ~1 micromolar H2S in buffer, so if we had >30 micromolar H2S circulating we would all smell like H2S if cut!)
b. I would also suggest inclusion of more contemporary references to general H2S donor approaches, examples could include: Pharmacol. Rev. 2017, 69, 497-564; Biochem. Pharmacol. 2018, 149, 110-123; Antioxid. Redox Signal. 2020, 32, 96-109.
c. In the discussion of EC-34 stability to Fenton chemistry, the basic experimental details should be included in the body of the manuscript. Things like concentrations, reaction times, etc are key for evaluating these experiments, so they should be stated in the text.
d. In the H2S release studies – again the concentrations, etc need to be in the body of the paper. It would also be very helpful if the y-axis were converted to H2S concentration rather than just pA. What is the efficiency of release? This should be measured using a calibration curve.
e. In Figure 4, showing the possible reaction pathways. Pathway A shows ester cleavage to form an intermediate thioacid, which should be hydrolyzed to release H2S. This makes sense. Pathway B shows formation of a terminal thiol with the authors state will release H2S. I’m not following how this is possible chemically. Do the authors have any evidence for this? There have not been donors that have terminal thiols that spontaneously release H2S that I am aware of, so the authors really need to back up this proposed reaction pathway.
Author Response
Reviewer 4
- The authors state that “Results of Olson [18] showed that endogenous H2S concentrations are between 30 and 300 μM in the plasma and blood” and reference a paper over a decade old. These high levels of H2S in plasma and blood are incorrect, and the contemporary literature agrees that free H2S levels are below 1 micromolar. The authors need to update this introductory material. (As a quick check- the human nose can smell ~1 micromolar H2S in buffer, so if we had >30 micromolar H2S circulating we would all smell like H2S if cut!)
The reviewer is correct. Thus, we modified the sentence as the follows.
„Early results have shown that endogenous H2S concentrations are in micromolar range. However, later it has been shown that H2S concentration in mouse brain and liver is around 20 nM [23,24].”
- I would also suggest inclusion of more contemporary references to general H2S donor approaches, examples could include: Pharmacol. Rev. 2017, 69, 497-564; Biochem. Pharmacol. 2018, 149, 110-123; Antioxid. Redox Signal. 2020, 32, 96-109.
We incorporated the suggested references. Please, see the following section in the introduction:
However, accumulating number of evidence suggest that the biological effect of H2S is not limited to a single pathway and a tissue type. It modulates different cellular targets in a cell and tissue dependent manner [11]. H2S can interfere with central metals, it can act as antioxidant coping with ROS/RNS and via S-persulfidation, it influences the function of different proteins [12].
As an example, H2S possess neuroprotective and cardioprotective effects [13-15]. Furthermore, it has been suggested that downregulated endogen H2S level plays a role in atherosclerosis; and exogenous H2S has therapeutic value against atherosclerosis via different mechanisms such as suppressing oxidative stress and inflammation, reducing endothelial dysfunction, platelet aggregation, and regulating lipid metabolism [16]. Furthermore, reduced serum H2S levels have been observed in patients suffering from heart failure and coronary heart disease [17,18].
- In the discussion of EC-34 stability to Fenton chemistry, the basic experimental details should be included in the body of the manuscript. Things like concentrations, reaction times, etc are key for evaluating these experiments, so they should be stated in the text.
We included the requested information. Please, see the following section:
„In case of Fenton reaction each tube contained 20 mM FeCl3, 20 mM EDTA and 10 mM ascorbic acid. Reaction mixtures for blank contained 1% DMSO only without EV-34. The control tube contained all reagents and EV-34, except hydrogen peroxide which is responsible for initiation of the reaction. Fenton reaction mixtures were stirred for 30 min at room temperature. Synthetic porphyrin reaction mixtures were contained acetonitrile, 100 mM formic acid, 10 mM Fe(III)-meso-tetra(4-sulfonatophenyl) porphine, EV-34 and hydrogen peroxide.”
- In the H2S release studies – again the concentrations, etc need to be in the body of the paper. It would also be very helpful if the y-axis were converted to H2S concentration rather than just pA. What is the efficiency of release? This should be measured using a calibration curve.
We changed the figure as it was requested.
- In Figure 4, showing the possible reaction pathways. Pathway A shows ester cleavage to form an intermediate thioacid, which should be hydrolyzed to release H2S. This makes sense. Pathway B shows formation of a terminal thiol with the authors state will release H2S. I’m not following how this is possible chemically. Do the authors have any evidence for this? There have not been donors that have terminal thiols that spontaneously release H2S that I am aware of, so the authors really need to back up this proposed reaction pathway.
In Path B we postulated the hydrolytic formation of intermediate 5, since compound 5 is a formaldehyde hemithioacylal. After hydrolysis of 5 hydroxymethanethiol could be formed, a hydrate of thioformaldehyde. Decomposition of this compound could lead either to the formation of formadehyde and hydrogen sulfide, or thioformaldehyde and water. Since we could not find any information in the literature on the decomposition of hydroxymethanethiol, therefore, for the sake of simplicity we removed the path B from the figure.
Round 2
Reviewer 4 Report
I have reviewed the responses from the authors with respect to the points raised by the initial reviewers. It appears that the authors have responded to these reviewer questions and suggestions well and that the manuscript is suitable for publication.
Author Response
Dear Reviewer!
Let us thank you for the time you devoted to review our manuscript and for the comments. We believed that your suggestions substantially increased the value of our manuscript.
Sincerely,
Istvan Lekli